# Clonal interference and genomic repair during strain coexistence in the gut

Nelson Frazão[1,2☯], Elsa Seixas[1☯], Manolo Mischler[1], Jorge Moura-de-Sousa[3], Hugo C. Barreto[1,4*], Isabel Gordo[1*]

1 Gulbenkian Institute for Molecular Medicine, Lisboa, Portugal, 2 Universidade Católica Portuguesa, Faculdade de Medicina, Centro de Investigação Interdisciplinar em Saúde, Lisboa, Portugal, 3 Institut Pasteur, Université Paris Cité, CNRS UMR3525, Microbial Evolutionary Genomics, Paris, France, 4 Université Paris Cité, CNRS, Inserm, Institut Cochin, Paris, France

☯ These authors contributed equally.
* isabel.gordo@gimm.pt (IG); hugo.barreto@inserm.fr (HCB)

## Abstract

Humans and other mammals are colonized by multiple strains of *Escherichia coli*, but the tempo and mode of evolution of different coexisting strains, between whom horizontal gene transfer (HGT) can occur, is poorly understood. Here, we follow in real time the evolution of two phylogenetic distinct strains of *E. coli* that co-colonize the mouse gut with different population sizes. We find qualitative differences in evolutionary dynamics between strains within the same host. In the strain with larger population size intense clonal interference occurs and polymorphism at a neutral marker locus is maintained, while in the strain with lower population size complete selective sweeps and loss of neutral marker polymorphism occurs. Strain coexistence is also accompanied by rich dynamics of HGT from one strain to the other. Strikingly, a rare HGT event could restore a previously lost genomic region in the recipient strain. Furthermore, we detect for the first time a case of phage piracy in the gut, where a putative phage satellite, lacking essential genes for their own replication, was likely mobilized by a helper phage to transfer between bacterial hosts. Our results show that HGT is a key mechanism underlying genetic exchanges and adaptive genomic repair in the mammalian gut.

## Author summary

*Escherichia coli* is a common species colonizing the intestine of many mammals including humans. Typically, more than one strain of this species colonizes the gut at a given time. While several works have studied adaptation of a single strain of *E. coli* within a host, our knowledge on how different strains that co-colonize the same mammalian host evolve is far from being well understood. Here we address this question using experimental evolution, where we can simultaneously follow the evolution of two distinct strains colonizing the intestine of

**Data availability statement:** Genome sequencing data has been deposited with links to BioProject accession number PRJNA1019258 in the NCBI BioProject database (https://www.ncbi.nlm.nih.gov/bioproject/). The code for the evolutionary model of accumulation of beneficial mutations is available on the GitHub platform at https://github.com/isabelgordo/HGTransferUnderStrainCoexistence. The custom R scripts and E. coli annotated genomes and plasmids are available on the GitHub platform at https://github.com/hugocbarreto/Lateral-gene-transfer-causes-genomic-repair-when-strains-coexist-in-the-gut.

**Funding:** N.F. was supported by the work contract (IGC-DL57NT-26), under the contract-program between Fundação para a Ciência e Tecnologia (FCT, https://www.fct.pt/) and Fundação Calouste Gulbenkian (FCG, https://gulbenkian.pt/). J.M.S. was supported by the Springboard to Independence grant (ANR LBX-62 IBEID AAP BOURSE S2I ROCHA) from the Laboratoire d'Excellence IBEID Integrative Biology of Emerging Infectious Diseases, Agence Nationale de la Recherche (ANR, https://anr.fr/). H.C.B. was supported by a cooperation agreement (CRC1310 - A6 awarded to I.G.) between the Gulbenkian Institute for Molecular Medicine (GIMM, https://gimm.pt/) and the University of Cologne (https://www.uni-koeln.de/) funded by Deutsche Forschungsgemeinschaft (DFG, https://www.dfg.de/), by the DREAM ANR-20-AMR-0002 grant from ANR, and by the HORIZON-MSCA-2023-PF-01 project number 101148351 - MICROINVADER, funded by the European Union (https://european-union.europa.eu/). This work was also supported by national funds from FCT (project reference PTDC/BIA-EVL/7546/2020, http://doi.org/10.54499/PTDC/BIA-EVL/7546/2020), and from the ERC-2022-ADG project number 101096203 - EvoInHi, funded by the European Union to I.G. Views and opinions expressed are however those of the authors only and do not necessarily reflect those of the European Union, the European Research Executive Agency, or the ERC. Neither the European Union nor the granting authority can be held responsible for them. The funders had no role in study design, data collection and analysis, decision to publish, or preparation of the manuscript.

**Competing interests:** The authors have declared that no competing interests exist.

the same host. We find differences in the evolutionary dynamics of each strain. Importantly, we also identify several events of genetic transfer from one strain to the other. Our results show that genetic transfer between two coexisting strains is a key evolutionary mechanism within the gut, as some of these events are highly adaptive and can lead to the genomic repair of a strain with genes from the other strain.

## Introduction

*Escherichia coli* is a highly diverse and versatile species, acting as a commensal bacteria of humans and other mammals [1], but also being implicated in several intra- and extra-intestinal diseases [2]. *E. coli* polymorphism for gene number and mobile genetic elements (MGEs) contributes to its extensive phenotypic diversity as a commensal and a pathogen [3]. A healthy human can be colonized by one dominant strain or, more commonly, by several strains [4]. The turnover of *E. coli* strains is also typically observed in the human gut, yet the drivers of such population structure dynamics are still not well understood. Transmission across human hosts, including mother-to-child [5], between animals and humans [6], as well as the import of bacteria from the environment, are thought to be important contributors to residence times of the strains in the human gut [7]. Yet, the transmission into a host gut is not the sole process contributing to *E. coli* within-host population structure. The evolutionary processes of mutation and horizontal gene transfer (HGT) are pervasive in *E. coli,* and both of them are known to contribute to within-host adaptation [8–10]. Thus, the speed of evolutionary adaptation of a strain could be an important contributor to its residence time [11]. In our previous work we found that when an invader *E. coli* strain was introduced in the gut environment of mice it was able to coexist with a *E. coli* resident already present in their microbiota. Furthermore, events of horizontal gene transfer (HGT) from the resident strain to the invader strain, driven by either bacteriophages or conjugative plasmids [12], could be detected. We also found that the acquisition of temperate bacteriophages, such as Rac-like prophages, by the invader strain conferred a metabolic advantage to the bacterial host [13,14]. However, the extent to which rates of adaptation vary across *E. coli* strains coexisting in the same host is still poorly known, as evolutionary dynamics are typically followed in a single colonizer strain. This was also the case in our previous work where one the *E. coli* strains was already present in the mouse gut [12,14]. Here, we take an *in vivo* experimental evolution approach of controlled co-colonization, using those two *E. coli* strains, enabling a direct comparison of each strain evolutionary dynamics.

As described previously [12,14], one strain is a natural commensal of the mouse gut and thus is expected to be well adapted to this environment. The other strain is the commonly used laboratory strain K12, originally isolated from the human gut and since then passaged in simple laboratory environments, accumulating several mutational changes in its genome [15]. We measured the accumulation of mutations after ~1600 generations (three months) of evolution and genetic transfers between

the strains. We find that despite the differences in strain origin, and significant differences in population sizes observed in fecal samples, both strains accumulate mutations at a similar pace in the gut. However, the dynamics of mutation accumulation are distinct between the strains: in the strain K12, with lower population size and evidence for HGT, adaptive mutations emerge and fix in that lineage, while in the mouse commensal strain, with larger population size, adaptive mutations emerge but stay polymorphic within the lineage. Importantly, we find a rare but remarkable case of genetic exchange between the coexisting strains. This led to the acquisition of novel DNA by the K12 strain, including the gain of an important gene, previously lost in this strain during laboratory propagation, which was strongly selected for during its adaptation to gut colonization.

## Results and discussion

### Coexistence of *E. coli* strains in the mouse gut and strain dependent mode of evolution

We used a classical model for *E. coli* colonization of the mouse gut, the streptomycin treated mouse model [16]. This treatment allows for the successful colonization by *E. coli* of the mouse gut while maintaining a complex microbiota composition [12], although with a reduced species diversity [17]. Briefly, female mice were simultaneously colonized with two *E. coli* strains (n = 9) (**Fig 1A**) after seven days of antibiotic treatment. One *E. coli* strain belongs to phylogenetic group B1 and is a common resident of the mouse gut [14]. Previous studies [12,18] indicate that this strain is a good donor of MGEs to the other strain used in this study, *E. coli* strain K12. The latter belongs to phylogenetic group A, is the best characterized *E. coli* strain and a good recipient of MGEs from the donor strain. Its genome is smaller than that of mouse resident strain. We will refer to the mouse commensal strain as "strain B1" and to K12 as "strain A". Each *E. coli* strain is labelled with two different neutral fluorescent markers, and different antibiotic resistances, to facilitate tracking the emergence of adaptive events in each strain during colonization (**Fig 1A**). In a clonal population, if adaptation is dominated by strong selective sweeps, then when an unconditionally beneficial mutation arises in a given fluorescent background and spreads to fixation (complete sweep) it will drag along the linked neutral fluorescent marker, which will also be fixed (genetic hitchhiking [19]). A more complex scenario can also occur if adaptation is dominated by clonal interference. This will happen when during the spread of a given beneficial mutation another beneficial mutation arises and escapes genetic drift. In this scenario, the frequency of the linked neutral fluorescent markers will fluctuate according to the dynamics of such competing clones. This may result in partial sweeps and maintenance of polymorphism of the neutral fluorescent markers for long periods of time [20]. Another scenario where neutral fluorescent marker polymorphism is expected to be maintained is if adaptation to the gut is characterized by mutations which are beneficial when rare but deleterious when at high frequencies (negative-frequency dependent selection) [21]. In line with our previous observations when these strains coexist in the mouse gut, we also expect the occurrence of events of HGT [12]. This additional mechanism of evolution has the potential to create new evolutionary events (e.g., due to phage transduction and lysogeny events, or transfer of plasmids by conjugation) and potentially introduce new selective pressures (e.g., due to prophage induction and cell lysis or to plasmid costs). Such events of horizontal transfer can further influence the dynamics of polymorphism at a neutral fluorescent marker.

Temporal series data of the abundance of each *E. coli* strain in the co-colonized mice shows that both strains can coexist for at least three months in the mouse gut (coexistence observed in eight out of nine mice, strain A failed to colonize in mouse 9, Figs 1B and S1, and S1 Table). Under coexistence, the absolute abundance of strain B1 in the feces was significantly higher than that of strain A (mean $Log_{10}$CFUs of strain B1: 8.5 (±0.4, S.E.) g$^{-1}$ of feces and mean $Log_{10}$CFUs of strain A 7.4 (±0.3, S.E.) g$^{-1}$ of feces, Linear mixed-effects model fit by maximum likelihood, $P = 0.0074$). Long-term maintenance of polymorphism at the fluorescent marker locus is observed in the strain with the larger population size (strain B1) (Fig 1B), despite strong changes in fluorescent marker frequency (S2 Fig). However, we observed loss of polymorphism at the neutral fluorescent marker in the strain with lower population size (Fig 1B and S1 Table). Thus, under strain coexistence in the mouse gut, the probability of maintaining polymorphism at a neutral locus is strain dependent (Binomial Test for proportions, $P = 0.00028$).

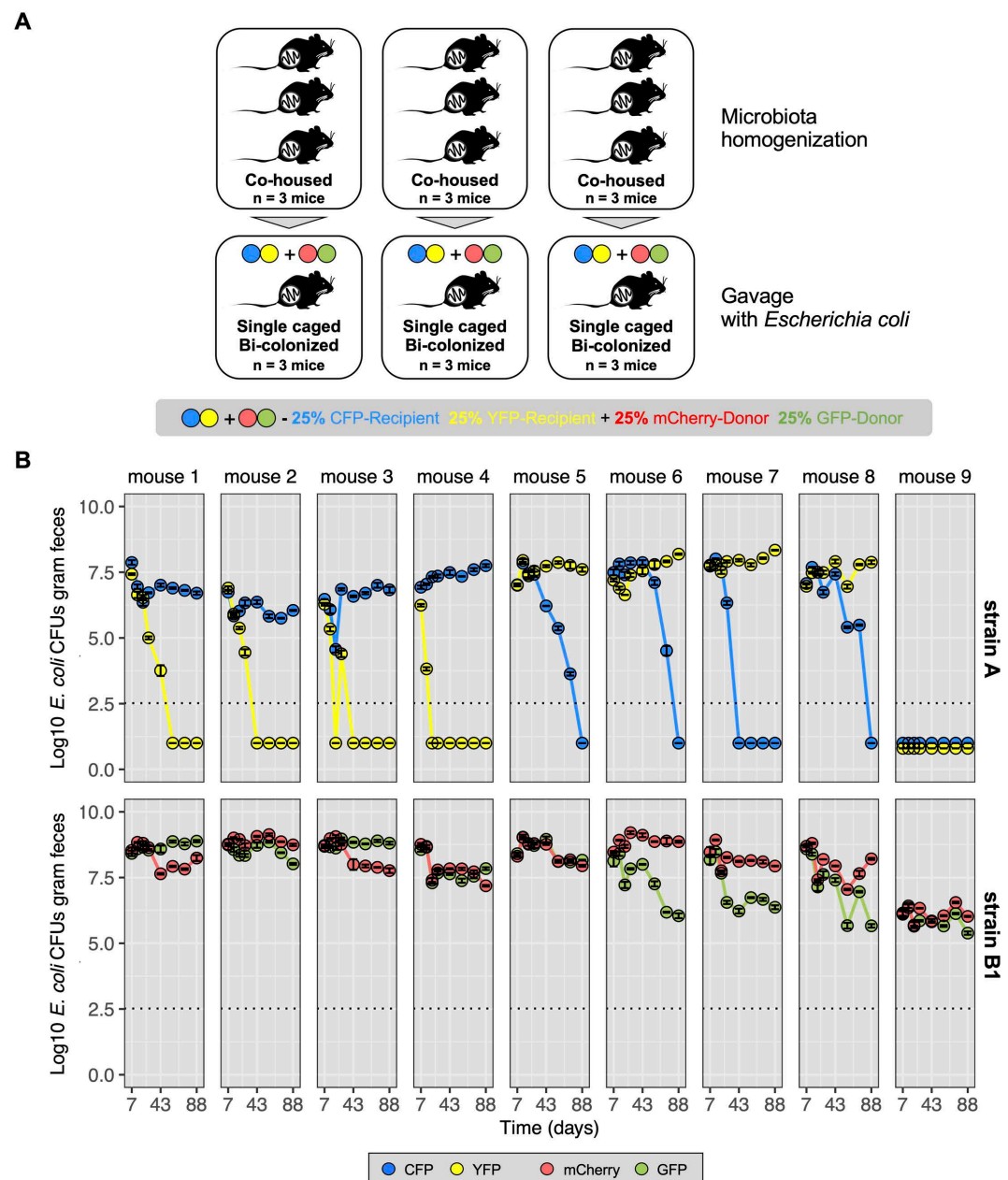

**Fig 1. Experimental design and distinct evolutionary dynamics between strains. (A)** Before co-colonization with *E. coli*, three cohorts of three mice (n = 3x3) were co-housed for 10 days to achieve microbiota homogenization: the first day without antibiotic treatment, followed by seven days of strep-tomycin treatment and two days without antibiotics in the water to clear the gut of any streptomycin traces [54]. In each cohort, mice were co-colonized with two fluorescent variants of strain A (CFP, blue circles; YFP, yellow circles) and two fluorescent variants of strain B1 (mCherry, red circles; GFP, green circles) *E. coli*, to follow their evolution in the mouse gut (see S1 Fig for more details). **(B)** Time series of the abundances of the two *E. coli* strains carrying one of the 4 fluorescent markers (CFP, YFP, mCherry and sfGFP). Dotted line represents the limit of detection for *E. coli*. Error bars represent the standard error of the mean (2*S.E.).

Given the large differences in the density of each strain and previous theoretical work suggesting that the probability of fixation of beneficial mutations in clonal populations depends on population size [22], we decided to explore conditions under which a model of positive directional selection in strictly clonal populations could explain the experimental data on strain coexistence and the maintenance or loss of polymorphism at a neutral locus. We performed simulations assuming a simple evolutionary model where adaptive mutations are unconditionally beneficial; the distribution of their effects (DFEBM) is similar between the strains and no gene transfer from one strain to another can occur. Simulations across a wide range of values for the mutation rate show that when the DFEBM is exponential (a commonly used assumption [23–29]), the model does not explain the experimentally observed data (see S3 Fig). However, the observed maintenance *vs* loss of neutral marker polymorphism in each strain could be seen under other DFEBMs, but only under particular conditions. Specifically, the simulations show that a model of positive directional selection is only compatible with the observed data (Fig 1B) if all beneficial mutations have similar effects (~1%) (S3 Fig).

## Genomic landscape of adaptive evolution of coexisting *E. coli* strains

To determine the evolutionary events that occurred in each *E. coli* strain under coexistence, and to quantify their rates of molecular evolution, we performed pool sequencing of clones from each strain after the 88 days of co-colonization (1584 generations, assuming 18 generations per day [21]).

In the mice where co-colonization was successful, the average number of mutation events detected in the strain B1 was higher than in strain A (10.3 ± 3.8 S.D. for strain B1 and 4.8 ± 1.6 S.D. for strain A, Paired T-test $P = 0.007$, S2 and S3 Tables). The 82 events that occurred in strain B1 were caused by SNPs (54 events, of which 34 are non-synonymous, four are synonymous and 16 intergenic), IS insertions (eight events), small insertions and deletions (nine events) and large deletions (11 events) (S3 Table). Across the eight mice where co-colonization was successful, the ratio of non-synonymous to synonymous mutations (dN/dS) was > 1, indicating adaptive evolution. Twelve targets of parallel evolution (*i.e.,* mutational events in the same target observed in different mice) were identified in strain B1 (Fig 2B and S3 Table), two of which correspond to large deletions. Consistent with a strong adaptive value, these deletions were detected in the majority of the mice (six and five out of eight co-colonized mice for the 87,305 bp and 85,944 bp deletions, respectively). Both deletions are flanked by an intact tRNA and a partial tRNA (S4 Table), suggesting that these regions are prone to recombination events. Interestingly, both deleted regions have homologous stretches of DNA (S4A Fig). Functionally, many genes within the deletions are predicted to code for integrases/transposases or to encode hypothetical proteins of unknown function, but a considerable portion of the deleted genes are predicted to be involved in virulence and metabolism (15% and 26% for the 87,305 bp and 85,944 bp deletions, respectively, S4B Fig). Despite the high level of parallel evolution observed in strain B1, none of the mutations detected in this strain reached 100%, which is consistent with the maintenance of both fluorescence markers in this strain (S2 Fig and S1 Table) and the absence of complete selective sweeps. On the other hand, evidence for competition between clones with different beneficial mutations is seen in the evolved populations of strain B1 in six out of eight co-colonized mice. In mouse 1 at least three different alleles at the *fimH* locus were detected (S3 Table), in mouse 2 three alleles occurred in the adhesin precursor YadA, mouse 3 shows two different SNPs at *dpiB* and in mouse 4 and mouse 6 two different mutations increased in frequency at a predicted membrane protein and in the transcriptional regulator *tdcA*, respectively. This data is consistent with a model of intense clonal interference in the strain with larger population size. Pool-sequencing of clones from an earlier time point (at day 43, *i.e.,* after 774 generations of evolution), shows the existence of more than one allele competing for fixation at loci *ydaA*, *dpiB*, *prokka01045* and *tdcA* (S3 Table), further corroborating that clonal interference shapes the selection dynamics of strain B1.

Molecular evolution of strain A was characterised by a total of 38 mutational events after 88 days of gut colonization (S2 Table). These were caused by SNPs (13 events, of which 10 are non-synonymous, two are synonymous and one is intergenic), IS insertions (11 events), small insertions and deletions (five events) and large deletions (nine events). We

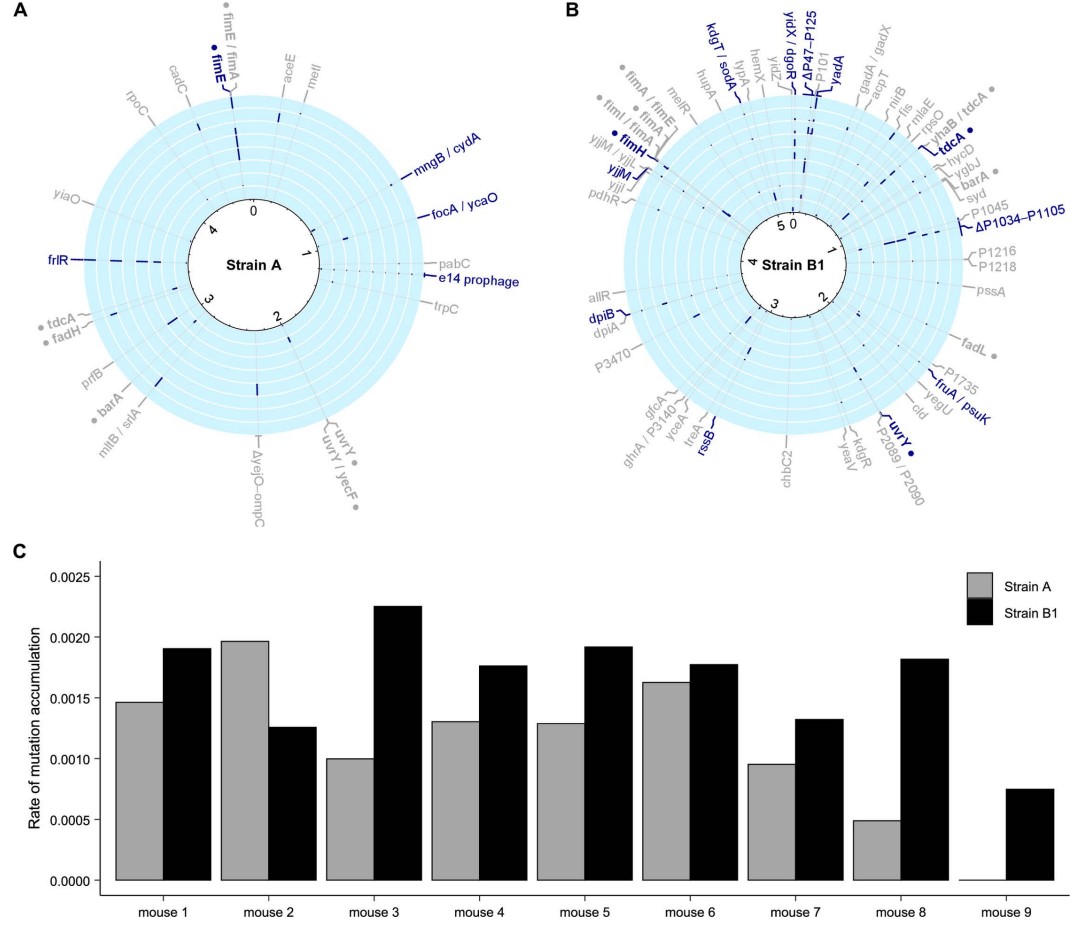

**Fig 2. Genetic basis of adaptation and rate of mutation accumulation in each *E. coli* strain.** Circular plots indicating the targets of evolution along the genome of strain A (**A**) and strain B1 (**B**) obtained from population sequencing at day 88. Targets that are considered adaptive within each strain, due to mutational parallelism, are highlighted in blue. Mutational events in the same gene or pathway observed in both strain A and strain B1, suggesting evolutionary convergence, are highlighted in bold and with the symbol •. Each circular track corresponds to one mouse, being the outermost ring the mouse 1 in both strains and the innermost ring the mouse 8 in strain A and mouse 9 in strain B1. The blue bars inside each circular track show the frequencies of each evolutionary event. Gene names with unknown function were renamed for clarity (*e.g.,* PROKKA_020290 is P20290 in the figure). The large deletions observed in strain B1 are indicated as ΔP47-P125 and ΔP1034-P1105, corresponding to the 87,305 bp and 85,944 bp deletions, respectively. **(C)** Rates of mutation accumulation per generation of each strain (strain B1 in black, strain A in grey) in each mouse. Rates were calculated by summing the allele frequencies of the evolutionary events detected in each strain after 88 days of colonization of the mouse gut. In mouse 9 the recipient strain failed to achieve persistent colonization.

detected five targets of parallel evolution in this strain across all the mice, which involved the following events: deletion of the cryptic prophage e14, mutations in the repressor of the fructoselysine operon *frlR*, IS insertions in the recombinase *fimE,* and in the intergenic regions *focA/ycaO* and *mngB/cydA* (Fig 2B and S2 Table). Consistent with their key role in adaptation to the gut, mutations in most of these targets were found in previous studies, where this strain colonized the gut as the sole *E.coli* strain [18,21,30–33], suggesting that these targets are independent of the co-colonization of strain A and B1. Across all co-colonized mice, nine selective sweeps (mutations that reached a frequency higher than 95% in an individual mouse) occurred in strain A. Interestingly, there were less detected partial sweeps in strain A, relative to strain B1, during the period in which evolution was followed. Multiple alleles segregating at the same locus were detected only in one out of eight mice for strain A *vs.* six out of eight for strain B1, after the 88 days of co-colonization. At an intermediate

point (43 days of co-colonization) partial sweeps were detected for strain A in two out of eight mice *vs.* four out of eight mice for strain B1. This suggests strain A experienced less clonal interference than strain B1 throughout the entire evolution period. The complete sweeps experienced by strain A can account for the loss of polymorphism at the neutral fluorescent marker in most of the mice. In mouse 1, a SNP in *flrR* reached 100% frequency, consistent with the fixation of the CFP fluorescent marker; in mouse 2, a 1 bp insertion in the intergenic region *mltB/srlA* likely caused the fixation of the CFP fluorescence; in mouse 3 a SNP in *frlR* likely caused the fixation of the CFP marker; in mouse 4 either a 22Kb deletion or a IS insertion caused the fixation of the marker; in mouse 5 either a 2 bp insertion in *frlR* or a IS insertion in *fimE* caused the fixation of the marker; and in mouse 6 a SNP in *prfB* likely caused the fixation of the YFP marker. In mice 7 and 8 no mutation event reached 100% frequency. This suggests that in these mice, HGT events could be invoked to explain the fixation of one of the markers (see below).

## Evolutionary convergence in *E. coli* strains during coexistence in the mouse gut

The strains used in our experimental evolution have distinct genome sizes, albeit in the same order of magnitude (strain A: $4.5 \times 10^6$; strain B:$5.2 \times 10^6$ base pairs), and some of the parallel adaptations that were detected within a given strain occurred in genes for which there was intra-strain variation. For example, from the eight mice in which co-colonization was successful, in five mice strain A evolved by accumulating mutations in *frlR*, a gene that does not exist in strain B1, and in five mice the strain B1 evolved by two large deletions of genes that are not present in strain A. Importantly, the extent to which co-existing *E. coli* strains will evolve by altering similar functions is still poorly understood. Our experimental design offers a unique opportunity to address this question. In the two phylogenetic distinct strains studied we found that several targets shared between strains underwent convergent evolution (*i.e.,* mutational events in the same gene or pathway observed in both strain A and strain B1). Evidence for evolutionary convergence occurred in nine genes or pathways (Fig 2A and 2B). A striking case was the *fim* operon, in which multiple alleles were detected in both strains (eight alleles in strain A and nine in strain B1) (Fig 2A and 2B). Consistent with an environment where different *E. coli* strains likely compete for niches and/or resources, the evolutionary convergent targets are related to motility/adhesion through type 1 fimbriae (*fimA, fimE, fimH, fimI*) [34], with these *fim* targets found in *E. coli* populations colonizing the same host (mouse 1 and 7). Convergent targets were also observed in central carbon metabolism (*barA* and *uvrY*) [35], fatty acid degradation (*fadH* and *fadL*) in mouse 2 [36], and threonine/serine metabolism (*tdcA*) [37].

## Rates of evolution in the mouse gut of phylogenetic distinct strains of *E. coli*

The different histories of each *E. coli* strain lead us to expect that their rates of molecular evolution should be very distinct. As strain B1 is a common resident of the mouse gut [14], it could be better adapted to this environment than strain A, which has been propagated in laboratory environments for decades. Thus, it could evolve at a slower pace. However, it is also possible that the environmental shift caused by the perturbation of the microbiome composition by antibiotic treatment could boost the rate of evolution in strain B1 [38]. Following Good *et al.* [39], and considering the genome sizes to be in the same order of magnitude, we estimated the genomic rate of mutation accumulation (M(t)) by summing the allele frequencies of all the mutations detected in each strain, after three months of co-colonization, and dividing this value by the number of elapsed generations (1584 generations, assuming 18 generations per day similar for each strain). For strain B1, M(t) was $1.8 \times 10^{-3}$ ($\pm 0.3 \times 10^{-3}$, S.D.) per generation, while for strain A it was $1.3 \times 10^{-3}$ ($\pm 0.5 \times 10^{-3}$, S.D.) per generation (Fig 2C). These estimates are not significantly different between strains (Paired T-test, $P = 0.07$), with the same observation when normalizing M(t) for each strain genome size (Paired T-test, $P = 0.23$), and are in contrast with the expectation given their history and respective mean abundances. In fact, no significant correlation was found between strain abundance and rate of evolution (Pearson's correlation, $P = 0.6$ for strain B1 and $P = 0.3$ for strain A, S5A Fig). Furthermore, lack of significant correlation was also observed with the gut microbiota diversity,

measured with by the Shannon Index (Pearson's correlation, $P = 0.8$ for strain B1 and $P = 0.5$ for strain A, S5B Fig) and the total number of amplicon sequence variants (ASVs) (Pearson's correlation, $P = 0.9$ for strain B1 and $P = 0.2$ for strain A, S5C Fig). These results support the observation that both strains evolve at a similar pace independently of their population size and the microbiota diversity in the gut environment. Interestingly, the rates of evolution estimated here are very similar to those observed in the initial period of evolution of the *in vitro* propagation of a *E. coli* strain in minimal media with glucose [39].

**Phenotypic evolution of *E. coli* strains evolved in the mouse gut**

We next assayed the phenotypic diversity of clones sampled from both strains, after they co-evolved for three months in the gut. We compared the growth of the evolved clones with that of their ancestor in media supplemented with mouse food (S6 Fig), an environment for which strain A (*E. coli* K12) is, *a priori*, less adapted to than strain B1 (a mouse commensal). Indeed, in these *in vitro* fitness assays strain A showed a strong signal of metabolic adaptation to the mouse diet (S5 Table), with their evolved clones showing a large fitness increase, both in terms of carrying capacity and maximum growth rate (Fig 3A and 3C). In contrast, for the strain B1 no clear signal of metabolic adaptation to the mouse diet was found in the evolved clones (Fig 3B and 3D). This result is consistent with the fact that strain B1 was isolated from the feces of mice that have experienced this diet for many generations.

**Horizontal gene transfer between strains of *E. coli* repairs an important K12 gene**

In a previous study where antibiotic treated mice were colonized solely with the K12 strain, we had observed that events of HGT could drive its evolution whenever another *E. coli* commensal residing in the gut would survive antibiotic treatment [12]. Here, where both strains simultaneously invade the gut microbiome, we find that, in most mice, the evolution of the *E. coli* K12 lineage was strongly impacted by several events of HGT, where DNA was acquired from the strain B1 (Fig 4A and S2 Table). In seven out of the eight co-colonized mice, these events were caused by the transfer of two prophages (Nef and KingRac) and two plasmids. While no mutations were detected in the plasmids acquired by strain A (S2 Table), in strain B1 plasmid copy number evolved and mutation events in the plasmids were observed (S3 Table).

Illumina sequencing of the *E. coli* populations of strain A and strain B1 at days 43 and 88 suggested a remarkable case of hybridization between coexisting strains in one of the mice (mouse 7). To validate this finding, we performed Nanopore long-read sequencing on two clones of strain A isolated from mouse 7 at day 88 (S6, S7 and S8 Tables), which confirmed the acquisition of a large genomic region in the chromosome of strain A. Specifically, a 93,712 bp region in strain A was replaced by a 120,546 bp region from strain B1 (Fig 4B). The strain A used to colonize the mice is a derivative of K12, and its original 93,712 bp region contains 99 genes (S6 Table), a cryptic prophage Rac, and a deletion of 13,756 bp adjacent to this cryptic prophage (when compared with the K12 reference genome NC_000913.2). In strain B1, the 120,546 bp region contains 135 genes, and is composed of an active prophage (named KingRac) and an adjacent region with the same genomic organization as that in strain A, with an average nucleotide identity (ANI) ranging from 80–100% for each 100 bp fragment. (Fig 4B and S6 Table). This event of DNA transfer led to the replacement of genes of strain A by homologous genes from strain B1, and also to the repair of the 13,756 bp region absent in strain A. This event of gene gain and repair is likely to have occurred *via* transduction (either specialized or lateral [40,41]) mediated by the prophage KingRac, given its proximity to this prophage in the genome. Importantly, the 13,756 bp region absent in strain A was originally composed of 13 genes (S6 Table) that have a role in key functions for *E. coli* gut colonization, and that got lost (*i.e.*, likely negatively selected) under *in vitro* propagation.

The functions repaired in strain A include: motility, adhesion, and biofilms (*uspE* [42] and *ydaM* [43]); stress response and DNA repair (*uspE* [42] and *ogt* [44]); metabolite transport (*abg* operon [45] and *ydaN* [46]); and the switch from aerobic to anaerobic metabolism (*fnr* [47]). The latter is particularly interesting as *fnr* is an important gene for the ability of

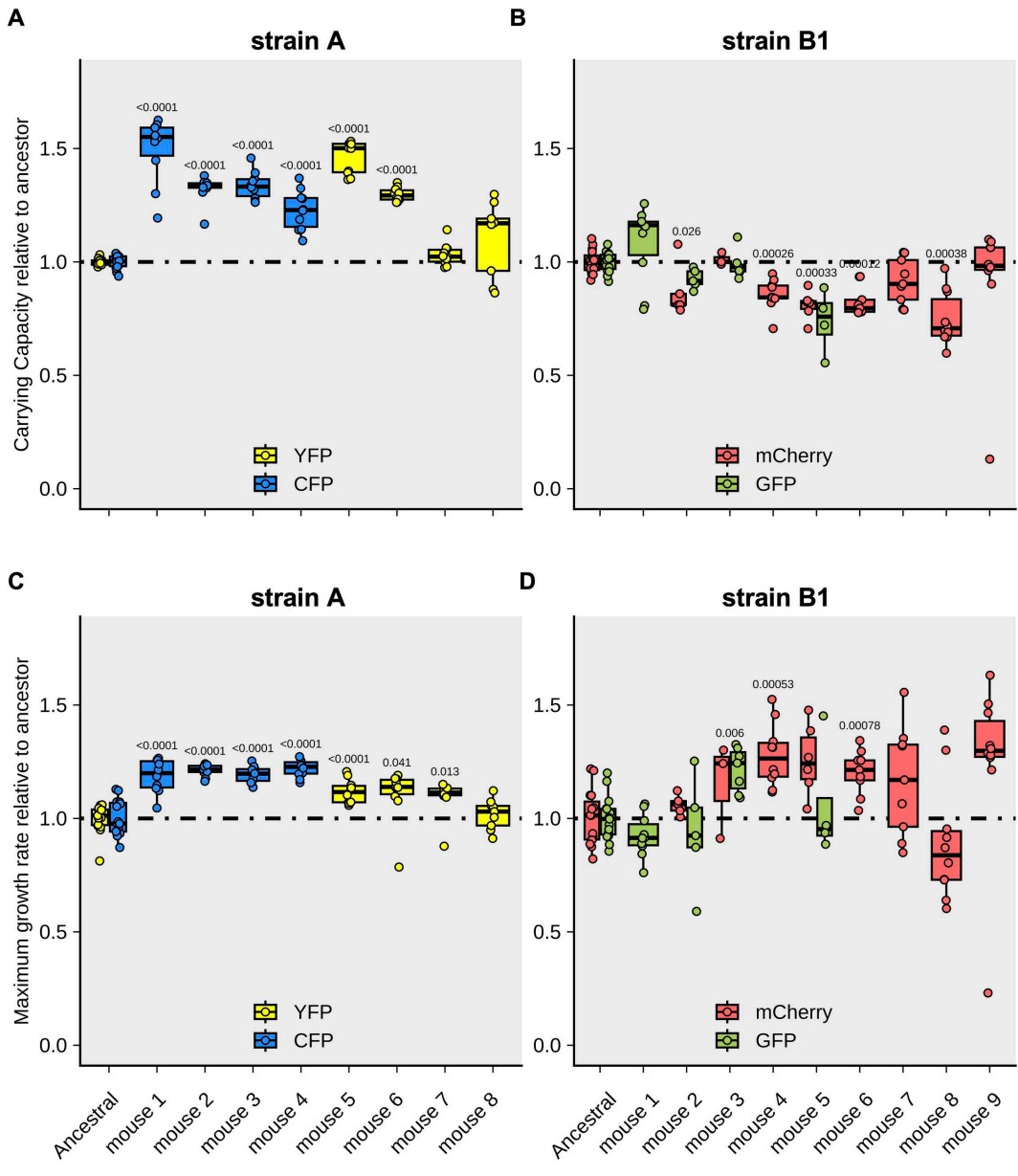

**Fig 3. Fitness assays *in vitro* of evolved clones when grown in mouse food. (A)** Relative maximum carrying capacity of clones of strain A sampled after three months of gut colonization of each mouse. **(B)** Relative maximum carrying capacity of clones of strain B1 sampled after three months of gut colonization of each mouse. **(C)** Relative maximum growth rate ($\mu/\mu_{anc}$) of clones of strain A sampled after three months of gut colonization of each mouse. The values of $\mu_{anc}$ (ancestor clones of the strain A) are 0.27 h$^{-1}$ (S.D. = 0.02) for the CFP-labeled clone and 0.28 h$^{-1}$ (S.D. = 0.02) for the YFP-labeled clone when grown in mouse food. **(D)** Relative maximum growth rate ($\mu/\mu_{anc}$) of clones of strain B1. The values of $\mu_{anc}$ for the ancestor strain B1 in mouse food are 1.08 h$^{-1}$ (S.D. = 0.10) for the sfGFP-labeled clone and 1.11 h$^{-1}$ (S.D. = 0.13) for the mCherry-labeled clone.

*E. coli* to colonize the mammalian gut. In fact, the presence of *fnr* in the genome of this bacterium is known to provide a fitness advantage in the mouse gut, as demonstrated by competitive fitness assays between *fnr* KO mutants and wild-type clones across different genomic backgrounds of *E. coli* [48]. In agreement with a fitness advantage conferred by this event, PCR typing revealed a steadily increase in the frequency of this event along time during the colonization of mouse

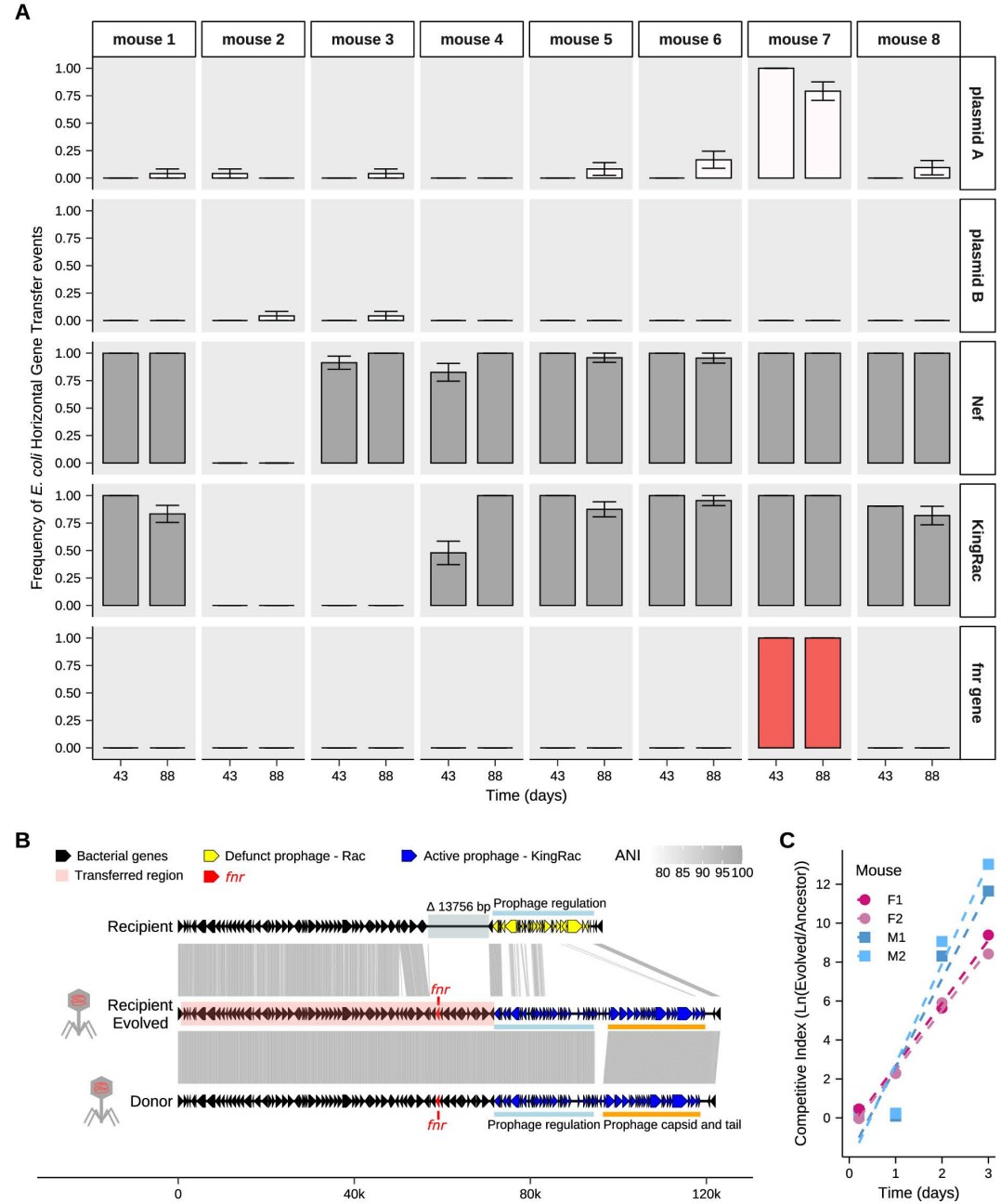

**Fig 4. Evolutionary dynamics of HGT from strain B1 to strain A. (A)** Frequency of *E. coli* HGT events from strains B1 to A at days 43 and 88 after colonization. Error bars represent the standard error of the mean (S.E.M). **(B)** Schematic representation of the massive number of genes gained by the strain A (recipient) from the strain B1 (donor), likely by transduction *via* the KingRac prophage. Within the genes gained is *fnr*, a core gene present in the majority of *E. coli* sequenced isolates. Gene arrow representation is to scale. Note that for the alignment the genomic orientation of the strain B1 is inversed. A phage symbol is shown next to the clones where the prophage is active. ANI, average nucleotide identity. **(C)** Competitive fitness assay of a clone evolved in mouse 7 (Evolved) that was competed against a double lysogen (with Nef and KingRac prophages) clone (Ancestor) in mice after one week of antibiotic pre-treatment. The y-axis shows the competitive index along time of colonization in mice (n = 2 male and n = 2 female). The selective coefficient, estimated from the slopes of the competitive index along time in each mouse, is 4.0 per day (S.E. = 0.5) or 0.22 per generation.

7 (Figs 4A and S7, and S2 Table). These results show that genes coding for functions not relevant outside the host can be regained due to HGT in the gut.

To directly measure the fitness benefit of a clone with this HGT event, we performed competitive fitness assays in new mice. We competed the evolved clone carrying the horizontally transferred *fnr* region against a lysogen of Nef and KingRac, given that these prophages confer an advantage by themselves [14]. We find a clear competitive fitness advantage of the evolved clone (Fig 4C and S9 Table), that together with the PCR typing (Fig 4A and S7 Fig) suggest a strong benefit for strain A driven by the genomic repair event.

Overall, our results show that HGT events of core bacterial DNA, even if rare are highly advantageous and able to restore in the gut previously lost genomic regions. We show here that this type of DNA transfer, can be detected in real-time during evolution in the mammalian intestine, underscoring its importance for the adaptation of *E. coli* to this environment.

### *In vivo* mobilization of a putative phage satellite

The analysis of the genomes of clones that evolved in mouse 7 identified the transfer, from strain B1 to strain A, of a genomic island predicted to correspond to an incomplete prophage (S8 and S10 Tables). Although it contains an integrase, this region lacks any genes – e.g., capsids or tails - that correspond to the assembly of viral structures (S11 Table). On the other hand, we observed the presence of genes that are typically associated with phage satellites, elements that are known to mobilize by exploiting co-infecting phages. Specifically the gene *ash*, which has a homologue in the P4 satellite and is a core gene of the P4-like family of satellites [49]. Additionally, the gene *alpA* was also detected, which is a core gene in four different phage satellite families [50]. This island also contains a gene homologous of TerS, that although often found in phages, it is also one of the core genes of phage inducible chromosomal islands (PICI) [51] (S8 Fig, S11 Table). We used SatelliteFinder [50] to understand if this genomic island corresponds to any known phage satellite. Although there were no reliable matches to any of the known families, SatelliteFinder identified this genomic island as an incomplete P4-like or PICI, due to homology with the core genes described above. This island might thus represent a potentially novel family of satellites, likely mobilized by Nef, KingRac, or both, but further work needs to be done to understand its identity and mobilization. In either case, to the best of our knowledge, this would correspond to the first documented transfer of a phage satellite *in vivo* in the mammalian gut.

### Conclusions

The potential for evolutionary change within bacterial species that colonize the mammalian gut is enormous, given the high population densities, large mutational inputs and the various selective pressures that occur in this ecosystem. Gene exchanges should also be a pervasive mechanism of evolution in the gut, given the high level of strain variation within each species, and the level of polymorphism of their MGEs. Using *E. coli* as a model for understanding the *tempo* and *mode* of evolution under strain coexistence, we show that strains with very distinct past histories and genome sizes stably co-exist in the gut and evolve under a strain-specific selection dynamics within the same host. Intra-strain convergent evolution was observed and large amounts of genetic information were exchanged from one strain to the other, *via* somewhat non-canonical HGT mechanisms, which can be caught within months *in vivo*. Such rare HGT events can be highly beneficial and can be captured as different co-existing strains co-evolve. Our results show that an unprecedented richness of evolutionary mechanisms and selection modes happens within the same host, when we follow the evolution of more than one strain, and that a typical gut resident strain can repair another strain that had experienced deleterious gene losses when outside the gut. Furthermore, our observations on convergent evolution during co-colonization and the large impact of HGT could be relevant for other bacterial species in natural populations [52,53]. Overall, our data highlights the complex combination of evolutionary forces that underly the genetic diversity of generalist and human commensal species such as *E. coli*.

## Materials and methods

### Ethics statement

This research project was ethically reviewed and approved by the Ethics Committee of the Instituto Gulbenkian de Ciência (license reference: A009.2018), and by the Portuguese National Entity that regulates the use of laboratory animals (DGAV - Direção Geral de Alimentação e Veterinária (license reference: 008958). All experiments conducted on animals followed the Portuguese (Decreto-Lei nº 113/2013) and European (Directive 2010/63/EU) legislations, concerning housing, husbandry and animal welfare.

### *Escherichia coli* strains

To facilitate the isolation from the mouse feces all the *E. coli* strains used in this study express fluorescent proteins and carry antibiotic resistance markers. The *E. coli* strain A express either a Yellow or a Cyan Fluorescent Protein, namely (YFP) or (CFP), respectively. Each one also carries a streptomycin resistance marker, as well as ampicillin (YFP) or chloramphenicol (CFP) resistance markers. Regarding the *E. coli* strain B1, these express either a Red (mCherry) or a Green Fluorescent Protein (sfGFP) and a chloramphenicol resistance marker. *E. coli* clones were grown at 37°C under aeration in liquid media Lysogeny Broth (LB) from SIGMA — or lactose supplemented McConkey agar plates and LB agar plates. Media were supplemented with antibiotics streptomycin (100 μg/mL), ampicillin (100 μg/mL) or chloramphenicol (30 μg/mL) when specified. Serial plating of 1X PBS dilutions of feces in LB agar plates supplemented with the appropriate antibiotics were incubated overnight and YFP, CFP, mCherry and sfGFP-labeled bacterial numbers were assessed by counting the fluorescent colonies using a fluorescent stereoscope (Zeiss Stereo Lumar V12). The detection limit for bacterial plating was ~300 CFU/g of feces [12]. To exclude that transfer of fluorescent markers between strains occurred, we examined the frequencies of strain-specific mutations unique to each fluorescent strain (S12 Table). The frequency of these mutations was consistent with their respective markers, suggesting that transfer of fluorescent markers between strains did not occur.

*In vivo* **evolution experiment.** We used a classical mouse gut colonization model where C57BL/6J mice (*Mus musculus*) were supplied by the Rodent Facility at Gulbenkian Institute. Animals (all female) were kept co-housed to homogenize the mouse microbiota during treatment with streptomycin (5 g/L) in the drinking water for seven days and the following two days, streptomycin treatment was absent to clean the gut from antibiotic traces [54]. This classical colonization model allows successful colonization with *E.* coli, maintaining a complex microbiota composition and eliminating the resident *E. coli*. Afterwards the animals were inoculated by gavage with 100 μL of an *E. coli* bacterial suspension of ~$10^8$ colony forming units (CFUs) and housed individually with drinking water not supplemented with streptomycin. Six- to eight-week-old C57BL/6J non-littermate female mice (n = 9) were kept in individually ventilated cages under specified pathogen free (SPF) barrier conditions at the animal facility. Mice were colonized by intragastrical gavage with two strains of *E. coli* (B1 and A). Fecal pellets were collected during approximately three months (88 days) and stored in 15% glycerol at -80°C for later analysis.

**Simulations of directional selection.** A simple evolutionary model of accumulation of beneficial mutations was simulated to enquire under which conditions the data obtained from measuring the abundance of neutral markers in each strain: the two fluorescent markers maintained in strain B1 and only one marker maintained in strain A. Beneficial mutations were assumed to be Poisson distributed with a constant mutation rate (u), similar to both strains, and the total population size of both strains was constant ($N = 10^6$). The model assumes non-overlapping generations, and each new generation is drawn *via* multinomial sampling of the precedent one, with probabilities proportional to the relative fitness of a mutant strain. The selective advantage of each new mutation was drawn from a fixed effect gamma distribution, and different shapes of this distribution were considered (shape = 1, corresponds to an exponential and shape = 100 is very similar to the case where all mutations have the same effect).

The total population is divided into four types (modeling the different fluorescence in the experiment, each with an initial equal fitness (set to 1), and the initial population sizes of each type were similar to those in the experiment: (0.05,0.05,0.45,0.45). 10 replicas for each parameter set were done. A large range of parameters was explored: selective advantage from $10^{-4}$ to $10^{-1}$ and mutation rate from $10^{-8}$ to $10^{-4}$, both with increments on a log scale. A Fisher's exact test was used to assess whether the difference between the simulated and the experimental results ones were not significantly different.

**DNA extraction for illumina sequencing.** DNA was extracted with Phenol-Chloroform [55] from *E. coli* populations (mixture of >1000 clones) from day 43 and day 88 after growing in LB plates supplemented with antibiotic to avoid contamination. DNA concentration and purity were quantified using Qubit and NanoDrop, respectively. The DNA library construction and sequencing were carried out by the Gulbenkian Institute Genomics facility using the Illumina Nextseq2000 platform.

**Bioinformatic analysis of sequenced evolved populations, evolved and ancestral clones.** References genomes for alignment of sequenced reads were K-12 substrain MG1655; Accession Number: NC_000913.2, for the alignment of reads from strain A; Accession Number: SAMN15163749 for the alignment of reads from strain B1; Accessions CP054663 and CP054664 for the two plasmids of strain B1 (with length 108557 bp and 68935 bp). Annotation of IS elements in the strain B1 genome was performed using ISEScan [56] (version 1.7.2.3), and only complete IS were considered for the annotation of the strain B1 reference genome. The annotated genomes for strain A, strain B1, and both strain B plasmids are available on the GitHub platform at https://github.com/hugocbarreto/Lateral-gene-transfer-causes-genomic-repair-when-strains-coexist-in-the-gut. Raw sequencing data were processed using fastp [57]. Sequencing adapters were removed, and raw reads were trimmed bidirectionally using 4 bp windows, retaining an average base quality of 20. Reads shorter than 100 bp and those with more than 50% of bases having a Phred score below 20 were removed, followed by base correction of overlapping reads and removal of duplicated reads. BBsplit [58] was used to remove reads with sequences highly divergent from the reference genome. Ancestral clones of strain A (CFP and YFP) and strain B1 (GFP and mCherry) were sequenced, and variant calling was performed against their respective reference genome using the 0.37.1 version of the BRESEQ pipeline [59] in clonal mode with default parameters. The CFP and YFP strain differed from Strain A reference genome by 13 and 12 mutations, respectively; the GFP and mCherry strains differed from the strain B1 reference by eight and seven mutations, respectively (S12 Table). For the evolved populations, variant calling was performed using the 0.37.1 version of the BRESEQ pipeline [59] with the polymorphism option on and default settings, except for: a) polymorphism minimum variant coverage of five reads; b) base quality cutoff of 30; c) minimum mapping quality of 20. To decrease the probability of detecting false positives in the evolved populations, sequencing reads from the ancestral clones were used as controls during variant calling. Variants predicted in the ancestral clones using the BRESEQ pipeline in polymorphism mode (using the parameters described above) were considered as false positives and excluded from the evolved population analysis. To further reduce the probability of detecting false positives in the evolved populations, variants in the ancestral clones that reached a frequency above 0.015 and were present in more than three reads were considered as false positives and removed from the evolved population analysis. Strain specific mutations (S12 Table) identified in the evolved population were used as a control for the fluorescent marker dynamics (S2 Fig). Variant calling on the samples from mouse 8 (day 43 and 88) for strain A and strain B1 revealed thousands of variants across the reference genome. Using BLASTN [60], we identified reads with 100% identity with genomes from *Enterobacter hormachei.* For these samples, BBSplit was run including genomes from *E. hormachei* (Accession Numbers: CP104691.1, CP104689.1, CP056649.1, CP044335.1, CP051132.1, CP116960.1, and CP019889.1). The seven genomes for *E. hormachei* were selected based on a 100% match of the contaminant reads with NCBI database. A similar pattern of thousands of variants across the reference genome of strain B1 was also observed in the samples from mouse 4 (day 43 and 88) and from mouse 6 (day 43). However, in this case BLASTN analysis revealed a 100% identity with the genome of strain A. For these samples, BBSplit was run including the reference genome of strain A (K-12 substrain MG1655;

Accession Number: NC_000913.2). Then, all samples were re-analysed using the BRESEQ pipeline as described above. To determine the frequency of prophage e14 excision in strain A, we first calculated the ratios between the e14 median coverage and the coverage in the left and right flank. The final frequency was obtained as the average of both ratios. We have previously shown that certain horizontal gene transfer (HGT) events can occur between strain B1 and strain A [12,18]. Specifically, the transfer of KingRac and Nef from strain B1 to strain A can be detected using the BRESEQ pipeline, as hundreds of variants confined to the genomic regions of the Rac and Qin cryptic prophages present in strain A. Importantly, the reads of these variants match 100% with the genome of strain B1, supporting HGT from strain B1 to strain A. To explore potential HGT events from strain B1 to strain A, and vice versa, in the evolved populations, we conducted BLASTN analyses of all reads aligning to the genomic positions of each variant against the reference genomes of both strains (Accession Numbers: NC_000913.2 and SAMN15163749). A variant was considered a potential HGT variant if more than 1% of the reads matched 100% with the non-equivalent reference genome (i.e., the reference genome of strain B1 for evolved populations of strain A, and vice versa). Potential HGT variants localized to the genomic regions of the Rac and Qin prophages was detected in the evolved populations of strain A of all mice except mouse 2, which did not exhibit potential HGT variants in these regions. Remarkably, the evolved populations from mouse 7 displayed a distinct pattern: numerous potential HGT variants extended into the non-prophage region flanking the Rac prophage were detected in this mouse (S7 Table). This suggests that a substantial genomic region was horizontally transferred from strain B1 to strain A. Importantly, this region included the *fnr* gene, essential for anaerobic respiration and known to contribute to increased fitness in the gut, which is absent in strain A but was restored in the evolved population of strain A from mouse 7. Nanopore sequencing of evolved strain A *fnr*-positive clones (see below) validated this observation (S8 Table), additionally confirming the replacement of a 93,712 bp region in strain A (from the first detected SNP to the last, including the cryptic Rac prophage) with a 120,546 bp region from strain B1 (from the first detected SNP to the last, including the KingRac prophage). No potential HGT variants were detected in the evolved populations of strain B1. All the remaining putative variants were verified manually in IGV [61] and the genome coverage for all populations was inspected for regions of missing coverage distinct from the ancestral clones. Manual inspection of genome coverage led to the identification of two large deletions in the strain B1 populations, and their frequency was calculated using a similar strategy as for the calculation of the excision of prophage e14. The custom R scripts used for this analysis are available at https://github.com/hugocbarreto/Lateral-gene-transfer-causes-genomic-repair-when-strains-coexist-in-the-gut).

**16S rRNA sequencing and microbiota analysis.** Fecal DNA was extracted from fecal samples of three different days (13, 27, and 87) from all the mice with a QIAamp DNA Stool MiniKit (Qiagen), according to the manufacturer's instructions and with an additional step of mechanical disruption [17]. 16S rRNA gene amplification and sequencing was carried out at Novogene Amplicon Metagenomic Sequencing Services, following the service protocol. For each sample, the V3-V4 regions of 16S rRNA genes were amplified using barcoded primers. The 16S rRNA V4 specific primers are 341F (5'-CCTAYGGGRBGCASCAG-3') and 806R (5'-GGACTACNNGGGTATCTAAT-3'). PCR reactions were carried out under the following cycling conditions: 98 °C for 1 min, 30 cycles of 98 °C for 10 s, 50 °C for 30 s, and 72 °C for 30 s, with an extension step of 72 °C for 5 min. Samples were then pair-end sequenced on Illumina NovaSeq 6000 platform, following Illumina recommendations. QIIME 2 version 2024.2 [62] was used to analyze the 16S rRNA sequences by following the authors' online tutorials (https://docs.qiime2.org/2024.2/). Briefly, the demultiplexed sequences were filtered using the "denoise-paired" command of DADA2 [63], and forward and reverse sequences were trimmed with the criteria allowing the highest number of reads kept (F-225, R-220). Microbiota diversity analysis was performed following the QIIME2 tutorial [62].

**Growth assays** in vitro. A pre-culture was prepared by inoculating the previously isolated single clones from the frozen stock to 150 μl MM + glucose [0.5 mM] in a 96-well plate, followed by incubation for 24 h at 37°C with 600 rpm in a benchtop shaker (Thermo-Shaker PHMP-4, Grant). To generate single clone growth curves the wells of a BioscreenC Honeycomb plate (Thermo Scientific) were filled with 295 μl culture medium and inoculated with 5 μl of pre-culture

PLOS Genetics

diluted in PBS, aiming at $10^5$ cells. The OD600 was measured with the BioscreenC, (Oy Growth Curves Ab Ltd.) for 24 h at 37°C with reading intervals every 10 minutes under continuous shaking with medium amplitude. The culture medium from mouse food was prepared using autoclaved Rat and Mouse food No.3 (SDS Special Diets Service). 75 g pellets of mouse food per liter were dissolved in sterile ddH$_2$O, then centrifuged and the supernatant filtered through a 0.22 µm PES membrane (Avantor, VWR). For each clone a maximum growth rate was estimated from 5 measurements of optical density (OD) during the exponential phase of the growth, by calculating the slope of Ln(OD – OD$_{blank}$) in a linear regression, which led to the highest growth rate and a $R^2 > 0.9$.

**Competitive fitness assay** in vivo. The relative fitness of an evolved *E. coli* clone of strain A, bearing the Nef and KingRac prophages, as well as the horizontally transferred *fnr* gene region (clone 7, S9 Table), was measured in the guts of mice (n = 4). Although the evolution experiment used only female mice, the competitive fitness assay included two female and two male mice to test for a sex independent fitness advantage. The evolved *E. coli* clone of strain A, expressing a yellow fluorescent protein (YFP), was competed against an ancestral *E. coli* clone with only Nef and KingRac prophages [14], expressing a cyan fluorescent protein (CFP). Both clones were grown overnight in 5 ml of Lysogeny Broth medium (LB) supplemented with streptomycin (100 µg/ml) from frozen stocks. The day after, bacterial density was adjusted (OD600 = 2), and a mixture of the two clones (1:1) was gavaged into the mice, which had been treated with streptomycin (5 g/L) in drinking water for the previous seven days, and without antibiotic treatment for two days previous to gavage. Fecal pellets were collected daily and frozen at -80°C for assessment of the abundances of each clone (YFP and CFP). The selective coefficient was estimated from the slopes of the competitive index ((Ln(YFP/CFP))over time in each mouse.

**PCR detection of ~69Kb (*repA*) and ~109Kb (*repB*) plasmids.** Specific primers for the amplification of *repA* and *repB* genes, were used to determine the frequency of the 68935 bp (~69 Kb) and 108557 bp (~109 Kb) plasmids, respectively, in the clones of strain A *E. coli* evolving populations at days 43 and 88 after gut colonization.

The primers used for *repA* gene were:

repA-Forward: 5'-CAGTCCCCTAAAGAATCGCCCC-3' and repA-Reverse: 5'-TGACCAGGAGCGGCACAATCGC-3'.

For *repB* the primer sequences were:

repB-Forward: 5'-GTGGATAAGTCGTCCGGTGAGC-3' and repB-Reverse: 5'-GTTCAAACAGGCGGGGATCGGC3'.

PCR amplification of plasmid-specific genes was performed in randomly isolated clones from the strain A evolved *E. coli* populations. PCR reactions were performed in a total volume of 25 µL, containing 1 µL of each clone growth in liquid LB media, 1X Taq polymerase buffer, 200 µM dNTPs, 0.2 µM of each primer and 1.25 U Taq polymerase. PCR reaction conditions: 95°C for 3 min, followed by 35 cycles of 95°C for 30 s, 65°C for 30 s and 72°C for 30 s, finalizing with 5 min at 72°C. DNA was visualized on a 2% agarose gel stained with GelRed and run at 160 V for 60 min.

**PCR detection of the Nef and KingRac prophages.** PCR amplification of phage-specific genes was performed as previously [14] to determine the frequency of lysogens in the strain A *E. coli* population at days 43 and 88 after gut colonization.

**PCR detection of the *fnr* gene.** Specific primers for the amplification of *fnr* gene, were used to determine its frequency in the strain A *E. coli* populations at days 14, 21, 28, 43 and 88 after gut colonization. The primers used were: *fnr*-Forward: 5'-CATTTAGCTGGCGACCTGGTGG-3', *ynaJ*_fw: 5'-TTCAGAGCAGACAACGGTGA-3' and *ttcA*_rv: 5'-GGGCGTATCGAGACGATGTT-3'. PCR amplification of the *fnr* gene was performed in randomly isolated clones from the strain A evolved *E. coli* populations. PCR reactions were performed in a total volume of 25 µL, containing 1 µL of each clone grown in liquid LB media, 1X Taq polymerase buffer, 200 µM dNTPs, 0.2 µM of each primer and 1.25 U Taq polymerase. PCR reaction conditions: 95°C for 3 min, followed by 35 cycles of 95°C for 30 s, 60°C for 30s and 72°C for 1 min, finalizing with 5 min at 72°C. DNA was visualized on 1% agarose gel stained with Xpert Green.

**Nanopore sequencing of *fnr*-positive clones and analysis.** DNA was extracted as described above from two *fnr*-positive clones of strain A (confirmation by PCR, see above), isolated from mouse 7 at day 88 post-colonization. DNA concentration and purity were accessed as described above, and DNA library construction and sequencing were carried out by the Gulbenkian Institute genomics facility using the Oxford Nanopore Tecnologies (ONT) and MinION. *De novo* assembly was performed using the Flye [64] pipeline (version 2.9.2) with the following parameters: a) ONT high-quality reads; b) estimated genome size of 4.7m; c) read error rate of 0.05. The assembly was then annotated with Prokka [65] using the annotated proteins from the reference genome of strain A (K-12 substrain MG1655; Accession Number: NC_000913.2) as trusted proteins to first annotate from. Putative prophage regions were previously identified using PHASTEST (S10 Table) and inspected manually for integration in the Nanopore sequenced strain A clones using IGV [61]. Mutations were identified using the 0.38.1 version of the BRESEQ pipeline [59] using the reference genome for strain A (K-12 substrain MG1655; Accession Number: NC_000913.2), with polymorphism option off and default settings except for a) base quality cutoff of 30; b) minimum mapping quality of 20. Average Nucleotide Identity (ANI) between the ancestral strain A, ancestral strain B1, and evolved strain A for the genomic region located adjacent to the prophage KingRac was calculated using fastANI [66] (version 1.33) with default parameters except: a) fragLen = 100.

**Annotation of the putative phage satellite.** The set of proteins in the region that corresponds to the genomic island, or putative phage satellite, transferred between strains B1 and A was first analysed using the Galaxy version of SatelliteFinder [50] (https://galaxy.pasteur.fr/root?tool_id=toolshed.pasteur.fr/repos/fmareuil/satellitefinder/SatelliteFinder/0.9) using the default parameters. The annotation of the individual proteins of the genomic island was based on the HMM profiles from PFAM (version 35.0) [67] and PHROG (version 4) [68]. The annotation of the proteins was performed using HMMER (version 3.3.2) [69], and by collecting all the hits (often >1 for the same protein) with a maximum e-value of $10-3$ and a minimum of 40% coverage as thresholds.

**Statistical analysis.** A Linear mixed model (R package nlme, v3.1 [70]) was used to analyze the load temporal dynamics of the *E. coli* lineages, while colonizing the mouse gut. A Binomial Test was used to compare strain polymorphism proportions. T-tests were used to compare differences in maximum growth rate between ancestor and evolved clones, and a paired T-test was used to compare rates of mutation accumulation of each strain. A P-value < 0.05 was considered for statistical significance.

## Supporting information

**S1 Fig. Mean abundance of B1 and A *E. coli* strains across time in each mouse gut.** Gut co-colonization with both *E. coli* strains occurs in mice 1–8. In mouse 9 strain A failed to colonize. Error bars represent the Standard Error (2*SE).
(TIF)

**S2 Fig. Frequency dynamics of the neutral fluorescent markers of each strain across approximately three months (88 days).** Gut co-colonization: frequency of the fluorescent markers in *E. coli* lineages of strain A (CFP, blue circles; YFP, yellow circles) and of strain B1 (mCherry, red circles; GFP, green circles) in the different mice (n = 9). Error bars represent the Standard Error (2*SE). In strain A, all mice exhibited fixation of one of the fluorescent markers, except for mouse 9, which was not successfully colonized. All the marker fixations can be attributed to a mutation reaching a frequency of 100%, except in the cases of mice 7 and 8. As no transfer of fluorescent markers or their linked resistance genes was observed, HGT events in other genomic regions must be invoked to explain neutral fluorescent marker fixation (S2 Table). In strain B1 there was maintenance of both fluorescence markers consistent with no selective sweeps observed (S3 Table).
(TIF)

**S3 Fig. Simulations of a simple model of directional selection under clonal evolution.** The heatmap shows the proportion of simulations compatible with the data of fluorescent marker abundances of each strain for each set of parameters (u and mean s). Data-compatibility is defined as maintaining only one marker in the strain with smaller population

size (strain A in the experiment) and the two markers in the strain with larger population size (strain B1 in the experiment). Parameters were chosen as follows: total population $N = 10^6$ (which is a rough estimate of the number of bacteria in a typical mouse fecal sample), initial proportion of clones of strain A $p = 0.1$, number of generation $g = 1600$ (88 days with 18 generation per day as in the experiment). The mutation rate u was tested across a large range of values, as well as the mean selective effect s of newly arising beneficial mutations. Mutations were assumed to have a (A) fixed effect or to follow a (B-D) gamma distribution. Three different shapes of the gamma distribution were considered: (B) shape = 100, which gives similar results as those in a model that assumes that all mutations have the same *s* value (fixed effect); (C) shape = 10; and (D) shape = 1, which corresponds to an exponential distribution. 10 replicates were simulated for each parameter set. A black star indicates that a significant number of them was compatible with the experimental results of the 8 mice (Fisher exact test). Globally, these simulations show that these simple models could only explain the observed data under very restrictive sets of parameters (mainly on the mean s of the selective effect distribution and on its variance, which has to be very small).
(TIF)

**S4 Fig. Genomic homology and gene functions of the two large deletions detected in strain B1.** (A) Schematic representation of the genomic regions with homology between the two large deletions observed in strain B1. (B) Relative frequency of the predicted functions of the genes present in the two large deletions observed in strain B1. ANI, average nucleotide identity. Functional categories were obtained from the annotation of Prokka.
(TIF)

**S5 Fig. Mutation rate does not correlate with average load and microbiota diversity.** Correlation between the genomic rate of mutation accumulation for each mouse and the (A) Log10 of the average load (day 7–88), (B) average Shannon Index (day 13, 27, and 87), and (C) average number of amplicon sequence variants (ASV) (day 13, 27, and 87). For all panels, a Pearson's correlation was performed.
(TIF)

**S6 Fig. Growth curves of ancestors and evolved clones in mouse food.** (A) Growth curves in media supplemented with mouse food of the ancestral (Control) and evolved clones of strain A sampled after three months of evolution inside the mouse gut. (B) Growth curves in media supplemented with mouse food of the ancestral (Control) and evolved clones of strain B1 sampled after 3 months of evolution inside the mouse gut.
(TIF)

**S7 Fig. HGT event causing a massive transfer of genes.** Dynamics of the spread of evolved clones which acquired a large lateral gene transfer event (which includes the *fnr* gene, typed here by colony PCR) located next to the prophage KingRac.
(TIF)

**S8 Fig. Sattelite genome map.** Scheme of the satellite genes and putative annotated functions.
(TIF)

**S1 Table. Loads of *Escherichia coli* along time in mice gut colonized with both strain A and strain B1 lineages.**
(XLSX)

**S2 Table. Whole genome sequencing of strain A *Escherichia coli* populations.** Mutation parallelism at corresponding day indicates the number of mice in which the exact same mutation was detected at day 43 or day 88. Gene parallelism at corresponding day indicates the number of mice in which mutational events targeting the same gene were detected at day 43 or day 88.
(XLSX)

**S3 Table. Whole genome sequencing of strain B1 *Escherichia coli* populations.** Mutation parallelism at corresponding day indicates the number of mice in which the exact same mutation was detected at day 43 or day 88. Gene parallelism at corresponding day indicates the number of mice in which mutational events targeting the same gene were detected at day 43 or day 88.
(XLSX)

**S4 Table. List of genes and predicted functions in the 87,305 bp and 85,944 bp deletions in strain B1.**
(XLSX)

**S5 Table. OD600 values of Escherichia coli clones growth curves.**
(XLSX)

**S6 Table. List of genes in the 93,712 bp region of *E. coli* K12 (NC_000913.2) and in the 120,546 bp region of strain B1.** In the list of genes that are not in prophage regions, empty lines mean that the gene is missing in the other *E. coli* strain.
(XLSX)

**S7 Table. List of mutations obtained from the Breseq output in strain A from mouse 7 at day 88 with potential HGT origin.** Bold indicates the region potentially transferred through KingRac transduction.
(XLSX)

**S8 Table. List of mutations obtained from the Breseq output for 2 fnr-positive clones from strain A from mouse 7 at day 88 after Nanopore sequencing.**
(XLSX)

**S9 Table. Competition of an *Escherichia coli* clone evolved in mouse 7 with a double lysogen clone with Nef and KingRac prophages.**
(XLSX)

**S10 Table. PHASTEST predicted prophages.**
(XLSX)

**S11 Table. Satellite genes and putative annotated functions.**
(XLSX)

**S12 Table. Mutations of Ancestral strain B1 and Ancestral strain A relative to reference genomes.**
(XLSX)

## Acknowledgments

We would like to thank the personnel of the GIMM's Rodent Facility, Genomic Facility and the Bioinformatics Unit for their assistance as well as Roberto Balbontín for the marked resident clone. We thank Daniela Güleresi for technical support in the experiments.

## Author contributions

**Conceptualization:** Nelson Frazão, Elsa Seixas, Hugo C. Barreto, Isabel Gordo.

**Data curation:** Nelson Frazão, Elsa Seixas, Jorge Moura-de-Sousa, Hugo C. Barreto, Isabel Gordo.

**Formal analysis:** Nelson Frazão, Elsa Seixas, Manolo Mischler, Jorge Moura-de-Sousa, Hugo C. Barreto, Isabel Gordo.

**Funding acquisition:** Isabel Gordo.

**Investigation:** Nelson Frazão, Elsa Seixas, Manolo Mischler, Jorge Moura-de-Sousa, Hugo C. Barreto.

**Methodology:** Nelson Frazão, Elsa Seixas, Manolo Mischler, Jorge Moura-de-Sousa, Hugo C. Barreto.

**Project administration:** Isabel Gordo.

**Resources:** Isabel Gordo.

**Supervision:** Isabel Gordo.

**Validation:** Nelson Frazão, Elsa Seixas, Hugo C. Barreto, Isabel Gordo.

**Visualization:** Nelson Frazão, Elsa Seixas, Manolo Mischler, Jorge Moura-de-Sousa, Hugo C. Barreto.

**Writing – original draft:** Nelson Frazão, Elsa Seixas, Jorge Moura-de-Sousa, Hugo C. Barreto, Isabel Gordo.

**Writing – review & editing:** Nelson Frazão, Elsa Seixas, Jorge Moura-de-Sousa, Hugo C. Barreto, Isabel Gordo.

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
