## [Decision Letter · Decision Letter 0]

PGENETICS-D-24-01403

Clonal Interference and genomic repair during strain coexistence in the gut

PLOS Genetics

Dear Dr. Barreto,

Thank you for submitting your manuscript to PLOS Genetics. After careful consideration, we feel that it has merit but does not fully meet PLOS Genetics's publication criteria as it currently stands. Therefore, we invite you to submit a revised version of the manuscript that addresses the points raised during the review process.

In particular, we wish to highlight a few points that the reviewers have raised that are important to address in a revision:

1. Reviewer 1 and 3 raised concerns about the simulation model proposed as well as interpretations about population size being a driving factor of the evolutionary dynamics observed. 

2. Reviewer 1 raises concerns about definitions and as a consequence interpretations about hard, soft, complete, partial sweeps. They also are concerned about confounding interpretations with alternative processes such as HGT and request some comparisons be made with mono-colonized populations (potentially from already published data). 

3. Reviewer 2 requests clarification about the pooled sequencing, as well as some thoughts on experimental vs natural communities. 

Please submit your revised manuscript within 60 days Apr 20 2025 11:59PM. If you will need more time than this to complete your revisions, please reply to this message or contact the journal office at plosgenetics@plos.org. Please include the following items when submitting your revised manuscript:

We look forward to receiving your revised manuscript.

Kind regards,

Nandita Garud

Academic Editor

PLOS Genetics

Justin Fay

Section Editor

PLOS Genetics

Aimée Dudley

Editor-in-Chief

PLOS Genetics

Anne Goriely

Editor-in-Chief

PLOS Genetics

**Journal Requirements:**

https://journals.plos.org/plosgenetics/s/submission-guidelines#loc-parts-of-a-submission

3) Please ensure all required sections (Abstract, Introduction, Results, Discussion, Methods) are present and in the correct order. Make sure section heading levels are clearly indicated in the manuscript text, and limit sub-sections to 3 heading levels. An outline of the required sections can be consulted in our submission guidelines here:

https://journals.plos.org/plosgenetics/s/submission-guidelines#loc-parts-of-a-submission 

5) We notice that your supplementary Figures are included in the manuscript file. Please remove them and upload them with the file type 'Supporting Information'. Please ensure that each Supporting Information file has a legend listed in the manuscript after the references list.

Potential Copyright Issues:

i) Figures 1A, and 4B. Please confirm whether you drew the images / clip-art within the figure panels by hand. If you did not draw the images, please provide (a) a link to the source of the images or icons and their license / terms of use; or (b) written permission from the copyright holder to publish the images or icons under our CC BY 4.0 license. Alternatively, you may replace the images with open source alternatives. See these open source resources you may use to replace images / clip-art:

7) Please amend your detailed Financial Disclosure statement. This is published with the article. It must therefore be completed in full sentences and contain the exact wording you wish to be published.

1)  If the funders had no role in your study, please state: "The funders had no role in study design, data collection and analysis, decision to publish, or preparation of the manuscript."

8) Please ensure that the funders and grant numbers match between the Financial Disclosure field and the Funding Information tab in your submission form. Note that the funders must be provided in the same order in both places as well. Currently, the order of the funders is different in both places.

Please indicate by return email the full and correct funding information for your study and confirm the order in which funding contributions should appear. Please be sure to indicate whether the funders played any role in the study design, data collection and analysis, decision to publish, or preparation of the manuscript.

**Reviewers' comments:**

Reviewer's Responses to Questions

**Comments to the Authors:**

**Please note that one of the reviews is uploaded as an attachment.**

Reviewer #1: Review uploaded as an attachment.

Reviewer #2: Understanding how different strains compete and coexist within a specific niche, as well as uncovering the molecular mechanisms driving such competition, represents a fascinating area of research. In the current manuscript, the authors address these compelling questions by analysing the competition and evolution of two E. coli strains in the gut environment. Overall, I find the manuscript highly engaging, particularly the sections where the authors describe several clear examples of horizontal gene transfer (HGT), including the transfer of large portions of the bacterial chromosome and a putative phage satellite. These findings provide valuable insights into microbial evolution and inter-strain interactions.

While I have no major criticisms, I offer the following suggestions to further improve the manuscript:

- Reorganisation of results. The initial part of the results, which outlines all the putative scenarios that might occur, detracts somewhat from the narrative. I suggest restructuring this section to first describe what actually occurs, followed by the corresponding explanations. The other hypothetical scenarios could be moved to the discussion section, where they might be better contextualised and explored.

- In the same initial section, I found that some of the terminology, such as "neutral polymorphism," may be challenging for non-experts to grasp. Providing a clearer explanation of these terms would enhance the readability and accessibility of the manuscript for a broader audience.

- If I understand correctly, the authors performed pooled sequencing of the strains at the end of the experiments. However, I missed seeing individual sequencing data for specific clones. It remains unclear whether all clones from the same strain exhibit identical mutations or whether different clones harbour distinct mutations. Including a more detailed analysis or discussion of clone-specific genetic variation would greatly enrich the understanding of strain dynamics.

- The manuscript mentions that mice were treated with antibiotics to establish colonisation. However, it is unclear how significantly this treatment reduces the microbiota and how natural the tested conditions are as a result. Additionally, it would be helpful to clarify whether other E. coli strains were present in the gut during the experiments and how they might influence the dynamics between the two focal strains.

- While I recognise it may be too late to conduct additional experiments, it would be valuable for the authors to discuss the potential outcomes of growing the two strains together in a controlled in vitro setting, such as in a nutrient-rich medium. Comparing the trajectories of strain competition and evolution under such conditions could provide further insights into the role of the gut environment in shaping the observed dynamics.

These suggestions aim to enhance the clarity, comprehensiveness, and impact of the manuscript. The study presents intriguing findings, and addressing these points will further strengthen its contribution to the field.

Reviewer #3: This manuscript presents the results of a co-colonization experiment of two E. coli strains in mice. The authors use fluorescent markers as a proxy for neutral genetic diversity in the two strains, one which is native to the mouse gut (B1) and the other the laboratory K12 strain (A). They track the evolution of both strains across 9 mice over approximately three months.

The authors find repeated mutations across mice in both strains, suggesting they are both undergoing adaptive evolution. They also observe several HGT events from B1 to A. The gene functions in these segments and competitive fitness assays suggest these HGT events were beneficial in A and restored functions that were previously lost in A.

The authors also characterize the rate of adaptation in the two strains and observe some interesting differences. Somewhat surprisingly, they detect a larger average number of mutations in the native gut strain B1 compared to the laboratory strain A (~10 vs ~5), which they attribute to the higher population size of the B1 strain, as measured by CFU/ml. However, they do not detected any difference in the rate of mutation accumulation in the two strains. In addition, they observe the fixation of one of the A fluorescent markers in all mice that successfully colonized, but no fixation in B1.

The results of the manuscript are new to my knowledge. The observation of convergent evolution during co-colonization and the large impact of HGT in the initial adaptation process is especially noteworthy and could be relevant for many other bacteria, including in natural populations [1, 2]. The manuscript is well-written and the results and methods clearly presented.

My only major comment has to do with the explanation for the loss of diversity in the A strain. Specifically, I was not convinced by the explanation of the authors that the lower population size of A would lead to hard sweeps but not in B1. First, do the authors have estimates of the population sizes of the strains in the mouse gut? Naively, I would expect the both strains to be in a high mutation supply regime (N\mu > 1). Also, even if the B1 population has more clonal interference, the fluorescent label should still be lost eventually, in which case there may be a quantitative but not qualitative difference between the two strains. It was also not clear to me how the rate of mutation accumulation in the two strains is similar but the average number of mutations in the strains differ by around a factor of 2.

Minor comments

- L. 114: I did not understand what is meant here by "much lower variance than an exponential." The variance of an exponential distribution is determined by its decay rate.

- L. 122: It would be useful if the authors could comment on the rate of increase in mutations compared to other measurements. For example, it seems like the numbers reported here are fairly similar to Lenski's LTEE.

- Fig. 1: The y-axis label is distorted.

References

[1] Sheppard, SK, et al., Convergence of Campylobacter Species: Implications for Bacterial Evolution, Science 320, 237-239 (2008).

[2] Birzu, G, et al., Hybridization breaks species barriers in long-term coevolution of a cyanobacterial population, bioRxiv:2023.06.06.543983 (2023)

**Have all data underlying the figures and results presented in the manuscript been provided?**

Reviewer #1: Yes

Reviewer #2: Yes

Reviewer #3: Yes

PLOS authors have the option to publish the peer review history of their article (what does this mean? ). If published, this will include your full peer review and any attached files.

**Do you want your identity to be public for this peer review?** For information about this choice, including consent withdrawal, please see our Privacy Policy .

Reviewer #1: No

Reviewer #2: No

Reviewer #3: No

**Figure resubmission:**
---

## [Decision Letter · Decision Letter 1]

PGENETICS-D-24-01403R1

Clonal Interference and genomic repair during strain coexistence in the gut

PLOS Genetics

Dear Dr. Barreto,

Thank you for submitting your manuscript to PLOS Genetics. After careful consideration, we feel that it has merit but does not fully meet PLOS Genetics's publication criteria as it currently stands. Therefore, we invite you to submit a revised version of the manuscript that addresses the points raised during the review process.

Please submit your revised manuscript within 30 days Jun 29 2025 11:59PM. If you will need more time than this to complete your revisions, please reply to this message or contact the journal office at plosgenetics@plos.org. Please include the following items when submitting your revised manuscript:

We look forward to receiving your revised manuscript.

Kind regards,

Nandita Garud

Academic Editor

PLOS Genetics

Justin Fay

Section Editor

PLOS Genetics

Aimée Dudley

Editor-in-Chief

PLOS Genetics

Anne Goriely

Editor-in-Chief

PLOS Genetics

**Additional Editor Comments:**

Dear authors

Thank you for the revision to the paper. As can be seen, the reviewers are satisfied with your work. However, Reviewer 1 raises two minor points -- if you would please address these points, we would be glad to accept the manuscript.

Sincerely,

Nandita Garud

**Reviewers' comments:**

Reviewer's Responses to Questions

**Comments to the Authors:**

Reviewer #1: We appreciate the author’s openness to our questions and critiques and their good-faith (and extensive) attempts to resolve them. We believe that the most recent revisions to the manuscript have significantly elevated the quality of the work.

However, there are several points we urge the authors to address prior to publication:

1. While the authors have successfully clarified that the simulations are meant to rule population size as a potential mechanism explaining differences in the evolutionary dynamics observed in populations of strains A and B1, respectively, we still question whether the simulations successfully achieve this aim. If the simulations ultimately are included in the main text: (1) the authors should explain why they heed simulations using models with exponentially distributed effects more than models which assign a uniform distribution of effects (S ~ 1%) to mutations, particularly as population size was able to explain differential dynamics observed in the empirical data under these latter models; and (2) the authors might want to consider running simulations in which mutations are modelled as purely neutral, as the loss of genetic diversity in the strain A populations would be consistent with genetic drift acting more rapidly/powerfully on smaller populations.

2. In response to our question about whether the M(t) statistic had been normalized by genome length, the authors added a line in the main text stating that they follow the approach of Good et al.—the study in which M(t) was originally developed, and in which the statistics was not normalized by genome length. However, we would like to point out that Good et al. applied this statistic to isogenic populations of E. coli that were allowed to evolve over time, and which therefore had the same genome length (at least they did at the beginning of the LTEE). This scenario differs from the study at hand, insofar as strains A and B1 had different genome lengths from the outset. That said, the authors may have decided not to normalize by genome length because the genome lengths of the two strains are similar enough to not qualitatively alter the M(t) results. If that is the case, we encourage the authors to say this explicitly, as it may allay readers’ concerns. It might also be advisable to communicate the actual size of each genome—if they are the same order of magnitude, it would further bolster the point that normalizing by genome size would not change the results.

3. Typo on line 360 (“geneticdiversity” should be “genetic diversity”).

Reviewer #2: This reviewer feels that the authors have adequately addressed all the questions I raised during my initial review. I would also like to note that I previously reviewed an earlier version of this manuscript when it was submitted to a different journal, and I find the current version to be significantly improved, both in terms of clarity and scientific content.

Reviewer #3: In the revision the authors have improved the presentation of the manuscript and clarified the potential sources of confusion regarding the interpretation of their results. In the process, they have addressed the questions raised in my previous review. I believe the manuscript will make a valuable contribution to the study of the role of HGT in bacterial evolution and have no further comments to add.

**Have all data underlying the figures and results presented in the manuscript been provided?**

Reviewer #1: Yes

Reviewer #2: Yes

Reviewer #3: Yes

PLOS authors have the option to publish the peer review history of their article (what does this mean? ). If published, this will include your full peer review and any attached files.

**Do you want your identity to be public for this peer review?** For information about this choice, including consent withdrawal, please see our Privacy Policy .

Reviewer #1: No

Reviewer #2: No

Reviewer #3: No

**Figure resubmission:**
---

## [Editor Report · Decision Letter 2]

Dear Dr Barreto,

We are pleased to inform you that your manuscript entitled "Clonal Interference and genomic repair during strain coexistence in the gut" has been editorially accepted for publication in PLOS Genetics. Congratulations!

Yours sincerely,

Nandita Garud

Academic Editor

PLOS Genetics

Justin Fay

Section Editor

PLOS Genetics

Aimée Dudley

Editor-in-Chief

PLOS Genetics

Anne Goriely

Editor-in-Chief

PLOS Genetics

**Data Deposition**

http://datadryad.org/submit?journalID=pgenetics&manu=PGENETICS-D-24-01403R2

**Press Queries**

---

## [Editor Report · Acceptance letter]

PGENETICS-D-24-01403R2

Clonal Interference and genomic repair during strain coexistence in the gut

Dear Dr Barreto,

We are pleased to inform you that your manuscript entitled "Clonal Interference and genomic repair during strain coexistence in the gut" has been formally accepted for publication in PLOS Genetics! Your manuscript is now with our production department and you will be notified of the publication date in due course.

With kind regards,

Zsofia Freund

PLOS Genetics

On behalf of:
